# Equalizing Seasonal Time Series Using Artificial Neural Networks in Predicting the Euro–Yuan Exchange Rate

**Marek Vochozka \*** , **Jakub Horák \*** and **Petr Šuleř**

The Institute of Technology and Business in České Budějovice, Okružní 517/10, 37001 České Budějovice, Czech Republic; petr.suler@cez.cz

\* Correspondence: vochozkam@email.cz (M.V.); horak@mail.vstecb.cz (J.H.); Tel.: +420-387-842-144 (M.V.); +420-380-070-218 (J.H.)

**Abstract:** The exchange rate is one of the most monitored economic variables reflecting the state of the economy in the long run, while affecting it significantly in the short run. However, prediction of the exchange rate is very complicated. In this contribution, for the purposes of predicting the exchange rate, artificial neural networks are used, which have brought quality and valuable results in a number of research programs. This contribution aims to propose a methodology for considering seasonal fluctuations in equalizing time series by means of artificial neural networks on the example of Euro and Chinese Yuan. For the analysis, data on the exchange rate of these currencies per period longer than 9 years are used (3303 input data in total). Regression by means of neural networks is carried out. There are two network sets generated, of which the second one focuses on the seasonal fluctuations. Before the experiment, it had seemed that there was no reason to include categorical variables in the calculation. The result, however, indicated that additional variables in the form of year, month, day in the month, and day in the week, in which the value was measured, have brought higher accuracy and order in equalizing of the time series.

**Keywords:** exchange rate; artificial neural networks; prediction; equalizing time series; seasonal fluctuations; categorical variable

## 1. Introduction

In this article, the authors focus on comparing the application of neural networks to the exchange rate using different modeling approaches. It has seemed before the experiment that there is no reason for including the categorical variables in order to capture the seasonal fluctuations of the two currencies exchange rates. Two sets of artificial neural networks were generated: (1) An independent variable was time, and a dependent variable was the Euro and Chinese Yuan exchange rate. (2) Seasonal fluctuation was represented by a categorical variable in the form of the year, month, day in the month, and day in the week in which the value was measured.

Predicting exchange rates is an important financial issue. In literature, great attention is paid namely to the difficulties in linear unpredictability of the exchange rate distribution. The development of the two currencies exchange rate can be predicted based on the statistical methods, causal methods, and intuitive methods. Conventional models based on the methods of linear time series often failed; therefore, non-linear time series models are used nowadays. In this article, the authors focus on comparing the statistical methods—artificial neural networks. This is a very interesting approach, which has not been solved in the scientific literature according to the available information. This question is very important for obtaining a model that enables to reduce the error predictions.

The objective of the contribution is to propose a methodology for considering seasonal fluctuations when equalizing time series by means of artificial neural networks on the example of Euro and Chinese Yuan. The article includes a literature review of the given issue, followed by a methodology part describing the methods and neural networks models used. The following sections deal with the results, where first the results without the involvement of seasonal fluctuations are presented, followed by the inclusion of a categorical variable. These two calculations are compared. In conclusion, the results are summarized and discussed.

## 2. Literature Review

The objective of the contribution is to propose a methodology for considering seasonal fluctuations when equalizing time series using artificial neural networks on the example of Euro and Chinese Yuan.

Time series can be described as certain observations of data, which are arranged in a horizon from the time perspective, from past to present. The analysis of time series is used mostly for future predictions (Mai et al. 2018). According to Rowland and Vrbka (2016), time series analysis is the area where neural networks are widely used. It is possible to use them for regression, classification, etc. Vrbka and Rowland (2017) stated that their main advantages were their capability of working with extensive data, the accuracy of results, and easy application in complex problems and forecasts. Vochozka and Vrbka (2019) claimed that numerical series were essential parts and important aspects of the whole process of exchange rate development. Numerical series are able to map past development and current status, and predict. Rodrigues et al. (2019) stated that by means of time series, it is possible to predict future values based on the past values. Accurate time series prediction is important for a wide range of areas. To determine the prediction methods and their time horizon, it is necessary to have the most accurate image of the prediction variables, nature of the data, and data availability, (Rostan and Rostan 2018).

Artificial neural networks provide powerful models for many economic classifications as well as for problems with regression. Neural networks are successfully used, e.g., for distinguishing between sound economic entities and those that are prone to bankruptcy, for predicting inflation–deflation, exchange rates, or shares price (Veselý 2011). In the last ten days, neural networks have changed from an esoteric tool in academic research into a common instrument, which helps auditors, investors, portfolio managers, and investment advisors in making financial decisions (Chen and Leung 2005). According to Klieštik (2013), artificial neural networks are computing models inspired by biological neural networks, which are used for a number of various areas. Figure 1 shows a model of an artificial neuron.

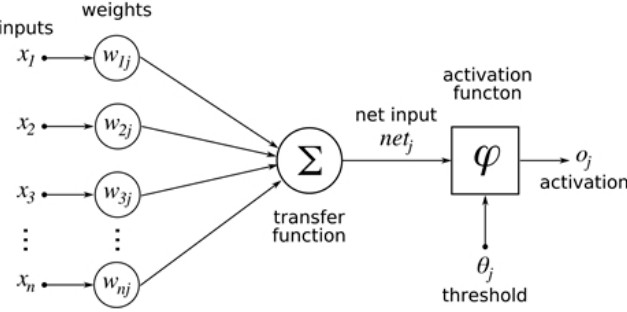

**Figure 1.** Artificial neuron. (Source: Own processing.).

Artificial neural networks are a reliable and flexible array of simple potent elements (neurons). This array enables interconnecting various inputs and outputs of neurons, favoring or suppressing some inputs, and minimizing the influence of a malfunctioning neuron on the overall result (Baptista et al. 2013).

Nowadays, neural networks are used namely for predicting values, solving potential problems in the future and are able to provide various solutions in estimating problems, regression, or optimization.

Martinovic et al. (2017) argued that the exchange rate was one of the most monitored economic variables. In the long run, it reflected the state of the economy, and in the short and medium term, it significantly affected it. The exchange rate can be monitored both in nominal and real terms (Jebran and Iqbal 2016). According to Clements et al. (2018), the link between the exchange rate and economic fundamentals formed a new basis. Based on the data situation analysis and empirical analysis of the business econometric model, it can be found out that the increasing export from China significantly affected the Euro market (Wang and Yang 2016).

According to Cai et al. (2012), CNY (Chinese Yuan) exchange rates can be considered as financial time series characterized by high non-linearity, uncertainty, and different behavior over time. Liu et al. (2009) focused on predicting exchange rates using RBF neural networks. Based on the experiments results, it can be stated that RBF neural networks are efficient and applicable for predicting the CNY exchange rates.

Nag and Mitra (2002) used a hybrid artificial intelligence method based on neural networks and a genetic algorithm. The results were appropriately compared with other non-linear statistical models, finding out that the hybrid method shows a higher performance than traditional non-linear time series methods. Galeshchuk and Mukherjee (2017) investigated the ability of neural networks to predict the changes in the forex rates EUR/USD, GBP/USD, and JPY/USD, finding out that the trained deep neural networks achieve a satisfactory accuracy of predictions. Similarly, Shen et al. (2015) came to the conclusion that the method proposed by them works better than traditional methods. The quality of the created models was also confirmed by Pradeepkumar and Ravi (2018), who studied and presented 82 such models for predicting exchange rates created between 1998 and 2017. They found that nearly all authors in this field demonstrated much better accuracy of hybrid models based on artificial intelligence.

Zhang and Wan (2007) designed a statistical fuzzy interval neural network with statistical interval input and output values for carrying out statistical fuzzy knowledge and predicting the exchange rate. The simulations were completed from the perspective of exchange rates between the American dollar and other three exchange rates (Japanese yen, British pound, and Hong Kong dollar). The results of the simulations proved that the neural network with a fuzzy interval provides great results of predictions.

Chen et al. (2009) used the integrated approach of an artificial neural network for predicting the American dollar and Chinese renminbi exchange rates. To achieve better predicting performance in terms of exchange rates, an integrated predicting model has been created, which uses an artificial neural network application called Alyuda Neuro Intelligence as a tool for predicting. The results of the study prove the efficiency and effectiveness of the artificial neural network integrated model and shall contribute to the development of this theory.

To characterize and predict implicated volatility on foreign exchange markets, Kearney et al. (2016) used a new technique of functional time series (FTS). It showed that the FTS model created realistic and credible implied shapes of volatility that corresponded to the empirical data in the volatility period between 2006 and 2013. Moreover, the FTS model outgrew the assumed predictions of volatility made by parametrical models.

Falát et al. (2015) examined the dynamics of EUR/GBP volatility dynamics using a statistical and neural approach, which was an alternative approach to time series modeling and predicting in economics. The objective of the contribution is to provide a method for modeling the dynamic economic time series. The authors propose an alternative approach to predictions of time series with an unstable volatility. They propose and implement several neural networks prediction models. The proposed models are realistic and acceptable for economic modeling.

Milovanovic et al. (2017) stated that endocrine neural networks (ENN) used artificial glands that allowed the network to adapt to external disorders. ENN is a network improved by creating a sensitivity parameter and implementation of orthogonal activation functions within a network structure. The network was tested using a series of experimental data in real time with the aim to predict the exchange rate of three widely used international currencies.

## 3. Materials and Methods

Data for analysis are available at the World Bank (2018) websites. For the purposes of the analysis, the information on the Euro to Yuan exchange rates is used. The time period for which the data are available is the period between 6th October 2009 and 21st October 2018. Daily exchange rates of both currencies are recorded. It is a total of 3303 input data. A unit is the number of Yuan for 1 Euro.

Data descriptive characteristics are shown in Table 1.

**Table 1.** Data characteristics.

| Samples | Case (Input Variable) | Euro to Chinese Yuan (Output Target) |
|---|---|---|
| Minimum (training) | 1.000 | 6.56070 |
| Maximum (training) | 3303.000 | 10.29420 |
| Mean (training) | 1643.786 | 8.08101 |
| Standard deviation (training) | 939.260 | 0.75919 |
| Minimum (testing) | 11.000 | 6.56980 |
| Maximum (testing) | 3302.000 | 10.25190 |
| Mean (testing) | 1664.707 | 8.06546 |
| Standard deviation (testing) | 957.966 | 0.76986 |
| Minimum (validation) | 20.000 | 6.58290 |
| Maximum (validation) | 3297.000 | 10.30830 |
| Mean (validation) | 1677.677 | 8.09958 |
| Standard deviation (validation) | 1438.254 | 1.20789 |
| Minimum (overall) | 1.000 | 6.56070 |
| Maximum (overall) | 3303.000 | 10.30830 |
| Mean (overall) | 1652.000 | 8.08147 |
| Standard deviation (overall) | 953.638 | 0.76206 |

Source: Own processing.

For data processing, DELL's Statistica software, version 12 was used. For calculation of the neural structures, the Data mining tool, namely neural networks (ANS—automated neural networks) was used. The above formulated automated neural networks, which the software contained, was used. Artificial neural network is a structure designed for distributed parallel data processing. It consists of artificial (or also formal) neurons, whose biological model is a neuron. Neurons are interconnected and transmit signals to each other and transform them with certain transmission functions. Neuron has a number of inputs, but only one output. The general neural network model is described as follows:

$$Y = S(\sum_{i=1}^{N}(w_i x_i) + \theta), \tag{1}$$

where $x_i$ are neuron inputs, $w_i$ are synaptic weights, $\theta$ is a threshold, $S_{(x)}$ is the neuron transfer function (activation function), and $Y$ is a neuron output.

Regression using neural structures was carried out. Due to the DELL´s Statistica software setting, we generated only multi-layer perceptron networks and radial basis function neural networks. Thus, other neural network types, such as recurrent neural networks (RNN), probabilistic neural networks (PNN), or Kohonen network (although used for different purposes) were not considered. MLP and RBF are the two most widely used types of neural networks that the software offers. MLP can be calculated as follows:

$$y_k^n = f(w_{0,k}^n + \sum_{i=1}^{m} y_i^{n-1} \times w_{i,k}^n). \tag{2}$$

The output of the *k*-th neuron is located in the *n*-th hidden or output layer. *f(x)* is the neuron transfer function, $w_{0,k}^n$ is the bias of the neuron, and *m* is the number of weights of the neuron.

MLP is a network consisting of several layers. The input layer, where the number of neurons corresponds to the size of the input vector, is followed by the hidden layer. The output layer, in which the number of neurons corresponds to the size of the output vector (see Figure 2) is connected to the input and hidden layers.

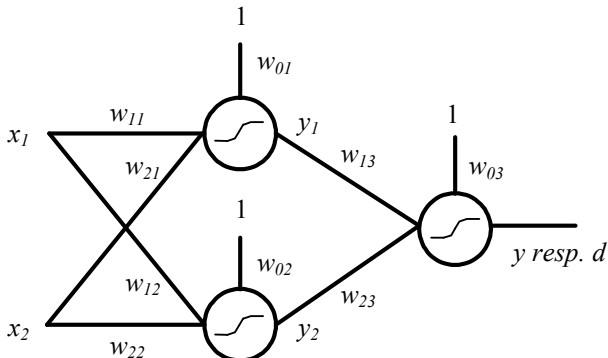

**Figure 2.** Graph representation of a general MLP network. (Source: Own processing.).

RBF can be calculated as follows:

$$f_k(x) = \sum_{j=1}^{k} w_i \varphi\left(\left|x - c_j\right|\right), \tag{3}$$

where $c_j$ is the point defining the center of $f_k(x)$ function and $\varphi$ specifies a particular type of radial base function.

Besides the input branching layer, RBF networks have two more layers of neurons—the hidden and output one. Those are characterized by a forward signal dissemination and learning with a teacher, while the learning speed is their great advantage. The RBF networks are used mainly for equalizing, approximating, and interpolating less structured problems with dispersed data. Graph representation of the network can be seen in Figure 3.

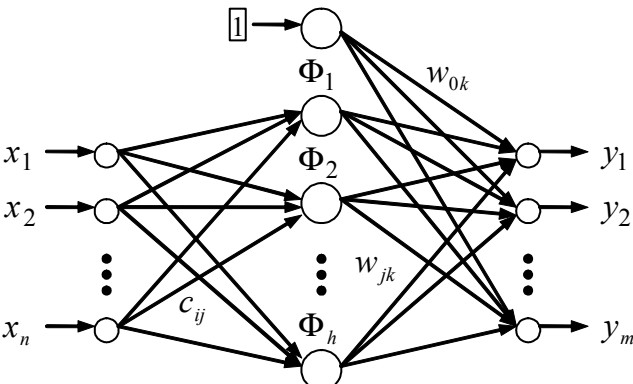

**Figure 3.** Graph representation of a RBF network. (Source: Own processing.)

Two sets of artificial neural networks were generated:

1.  An independent variable was time (it is a time trend). A dependent variable was the Euro and Chinese Yuan exchange rate.
2.  A continuous independent variable was time. Seasonal fluctuation was represented by a categorical variable in the form of the year, month, day in the month, and day in the week in which the value was measured. Each variable were examined separately. We thus worked with the possible daily, monthly, and annual seasonal fluctuations of the time series. A dependent variable was the Euro and Chinese Yuan exchange rate.

Then, we worked analogically with the data sets. The time series was divided into training, testing, and validation data sets. The first set contained 70% of the input data. Based on the training data set, neural structures were generated. The remaining two data sets contained 15% of the input data each. Both data sets were used for verification of the generated neural structure or model reliability.

The delay of the time series was 1. We generated 100,000 neural networks, out of which 5 with the best characteristics[1] were retained. The hidden layer of the multi-layer perceptron network contained at least two neurons (50 at most). In the case of the radial basis function, the hidden layer contained at least 21 neurons (30 at most). For multi-layer perceptron network, we considered the following distribution functions in the hidden and output layers:

- Linear,
- Logistic,
- Atanh,
- Exponential,
- Sinus.

Other settings were default (according to the ANN tool). Finally, the results of both groups of retained neural networks will be compared.

## 4. Results

This section was divided by two subheadings—neural structures A and neural structures B.

### 4.1. Neural Structures A

Based on the established procedure, 100,000 neural networks were generated, out of which 5 with the best characteristics were retained. Their overview is seen in Table 2.

**Table 2.** Overview of retained neural networks.

| Index | Network | Training Perform. | Testing Perform. | Validation Perform. | Training Error | Testing Error | Validation Error | Training Algorithm | Error Function | Activation of Hidden Layer | Output Activation Function |
|-------|---------|-------------------|------------------|---------------------|----------------|---------------|------------------|--------------------|----------------|---------------------------|---------------------------|
| 1 | RBF 1-24-1 | 0.983554 | 0.984770 | 0.984081 | 0.009370 | 0.009020 | 0.009390 | RBFT | Sum.sq. | Gaussian | Identity |
| 2 | RBF 1-29-1 | 0.981894 | 0.981738 | 0.983707 | 0.010306 | 0.010730 | 0.009532 | RBFT | Sum.sq.. | Gaussian | Identity |
| 3 | RBF 1-30-1 | 0.984826 | 0.985312 | 0.984546 | 0.008650 | 0.008742 | 0.009129 | RBFT | Sum.sq. | Gaussian | Identity |
| 4 | RBF 1-28-1 | 0.984362 | 0.984673 | 0.983238 | 0.008913 | 0.009007 | 0.009832 | RBFT | Sum.sq.. | Gaussian | Identity |
| 5 | RBF 1-26-1 | 0.983486 | 0.984695 | 0.984107 | 0.009407 | 0.009014 | 0.009311 | RBFT | Sum.sq. | Gaussian | Identity |

Source: Own processing.

These are only radial basis neural networks. The hidden layer contains only one variable—time. The hidden layers of the neural networks contain between 24 and 30 neurons. The output layer logically contains only one neuron and one output variable—the Euro to Yuan exchange rate. For all three networks, training algorithm RBFT was applied. For the activation of the hidden layer, all neural structures used the identical function, Gaussian function. Similarly, for the activation of the output layer of neurons, the same function (Identity) is used. For more details, see Table 2.

Training, testing, and validation performance was also interesting. Generally, we were looking for such a network whose performance in all data sets was ideally the same (here, it should be noted that the division of the data into the data sets was random). The error level was as low as possible.

The performance of the individual data sets is in the form of the correlation coefficient. The values of the individual data sets by specific neural networks are given in Table 3.

---

[1] The least squares method was used. Generating networks was finished if there was no improvement, i.e., the sum of the square was not lower. We thus retained the neural structures whose sum of residuals squares to development of Euro and Chinese Yuan was as low as possible (zero in the ideal case).

**Table 3.** Correlation coefficients of individual data sets.

| Network | Euro to Chinese Yuan (Training) | Euro to Chinese Yuan (Testing) | Euro to Chinese Yuan (Validation) |
|---|---|---|---|
| 1. RBF 1-24-1 | 0.983554 | 0.984770 | 0.984081 |
| 2. RBF 1-29-1 | 0.981894 | 0.981738 | 0.983707 |
| 3. RBF 1-30-1 | 0.984826 | 0.985312 | 0.984546 |
| 4. RBF 1-28-1 | 0.984362 | 0.984673 | 0.983238 |
| 5. RBF 1-26-1 | 0.983486 | 0.984695 | 0.984107 |

Source: Own processing.

It results from the table that the performance of all retained neural structures is almost the same. The slight differences do not have any influence on the performance of the individual networks. The value of the correlation coefficient of all training data sets is in the interval of more than 0.981 to more than 0.984. The correlation coefficient of the testing data sets achieves the same values as in the case of the training data set. The correlation coefficient of the validation data set of all neural networks is above 0.983. It should be also noted that the error for all data sets is in the interval between almost 0.009 to more than 0.01. The differences in the error of the equalized time series in the individual data sets are almost negligible.

To choose the most suitable neural structure, further analysis of the results obtained must be carried out. Table 4 shows the basic statistical characteristics of the individual data sets for all neural structures.

**Table 4.** Statistics of individual data sets by retained neural structures.

| Statistics | 1. RBF 1-24-1 | 2. RBF 1-29-1 | 3. RBF 1-30-1 | 4. RBF 1-28-1 | 5. RBF 1-26-1 |
|---|---|---|---|---|---|
| Minimal prediction (training) | 6.85480 | 6.70498 | 6.85756 | 6.86930 | 6.79547 |
| Maximal prediction (training) | 10.21765 | 9.92194 | 10.20493 | 10.14413 | 10.17991 |
| Minimal prediction (testing) | 6.85511 | 6.70617 | 6.85749 | 6.86959 | 6.79648 |
| Maximal prediction (testing) | 10.21718 | 9.92191 | 10.20358 | 10.14384 | 10.17832 |
| Minimal prediction (validation) | 6.85528 | 6.70502 | 6.85825 | 6.87002 | 6.79581 |
| Maximal prediction (validation) | 10.21006 | 9.92164 | 10.19128 | 10.14073 | 10.18000 |
| Minimal residuals (training) | −0.45897 | −0.70049 | −0.40403 | −0.39654 | −0.45631 |
| Maximal residuals (training) | 0.56285 | 0.41491 | 0.47297 | 0.49684 | 0.58830 |
| Minimal residuals (testing) | −0.47026 | −0.69190 | −0.38432 | −0.38399 | −0.43711 |
| Maximal residua (testing) | 0.47346 | 0.40493 | 0.43641 | 0.45880 | 0.41700 |
| Minimal residuals (validation) | −0.45710 | −0.56530 | −0.35886 | −0.38681 | −0.42207 |
| Maximal residuals (validation) | 0.57571 | 0.42783 | 0.46780 | 0.50829 | 0.58714 |
| Minimal standard residua (training) | −4.74158 | −6.90008 | −4.34425 | −4.20036 | −4.70472 |
| Maximal standard residuals (training) | 5.81469 | 4.08702 | 5.08551 | 5.26279 | 6.06549 |
| Minimal standard residuals (testing) | −4.95139 | −6.67959 | −4.11032 | −4.04606 | −4.60389 |
| Maximal standard residuals (testing) | 4.98511 | 3.90916 | 4.66740 | 4.83434 | 4.39212 |
| Minimal standard residuals (validation) | −4.71717 | −5.79008 | −3.75590 | −3.90102 | −4.37405 |
| Maximal standard residuals (validation) | 5.94113 | 4.38203 | 4.89604 | 5.12617 | 6.08474 |

Source: Own processing.

In ideal case, the individual statistics of the neural networks match horizontally in all data sets (minimum, maximum, residuals, etc.). In the case of the retained neural networks, the differences of the equalized time series are minimal both in terms of absolute values and residuals. Still, we are not able to identify unambiguously which of the retained neural networks shows the best or the most suitable results. All the structures appear to be applicable in practice.

Figure 4 shows a line graph that represents the actual development of the Euro and Yuan exchange rate and also the development of predictions by means of the individual generated and retained networks (or equalized time series). The blue curve follows the actual development of the exchange rate, while the other curves always follow one of the retained neural networks.

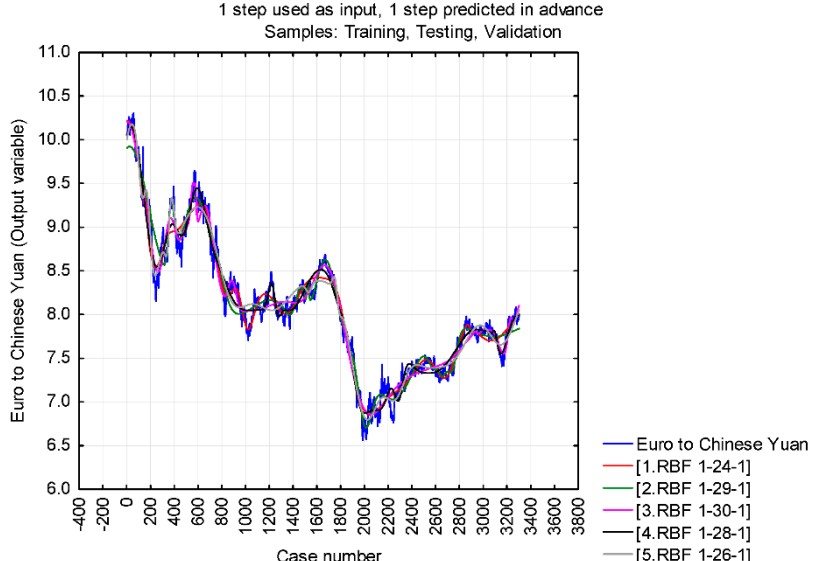

**Figure 4.** Line graph—development of the Euro to Yuan exchange rate predicted by neural networks in comparison with the actual exchange rate in the monitored period. (Source: Own processing.)

It follows from the graph that all the neural networks predict the development of the Euro to Yuan exchange rate slightly differently in the individual intervals. However, what is important is not the similarity of the individual network predictions but their similarity to (or degree of conformity with) the actual development of both currencies exchange rate. Even in this respect, it may be stated that all the retained neural networks appear to be very interesting at first sight. They follow the basic gradient of the curve representing the development of Euro to Yuan exchange rate and at the same time perceive the extremes of this curve.

Since the graph in Figure 4 contains 3303 data on Euro exchange rate to Yuan, it may appear unclear. Therefore, it would be good to demonstrate the situation in a selected data interval. The graph in Figure 5 shows a comparison of the actual development of the Euro to Yuan exchange rate in the interval of the last 100 days of the monitored period, that is, from 14th July to 21st October 2018.

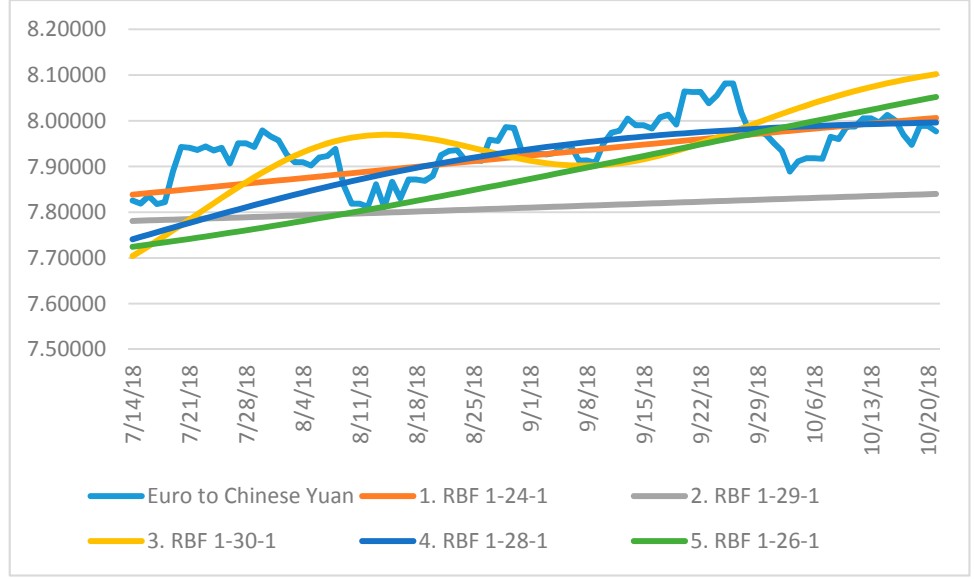

**Figure 5.** Line graph—development of the Euro to Yuan exchange rate predicted by neural networks compared to the actual exchange rate between 14th July and 21st October 2018. (Source: Own processing.)

It is evident from the graph that neither of the retained neural networks is able to imitate the actual development of Euro to Yuan in the monitored interval. The closest to the actual development is network 1. RBF 1-24-1. At the beginning of the monitored period, it almost coincides with the actual price of Yuan; similarly, at the end of the monitored period, it is not much different from the target value. In both cases, the difference is of the order of a few hundredths. Even the least accurate network, 3. RBF 1-30-1, differs from the actual exchange rate by less than 0.15 Yuan. Therefore, examining the residuals also appears to be interesting. The development of residuals between 14th July and 21st October 2018 is shown in Figure 6.

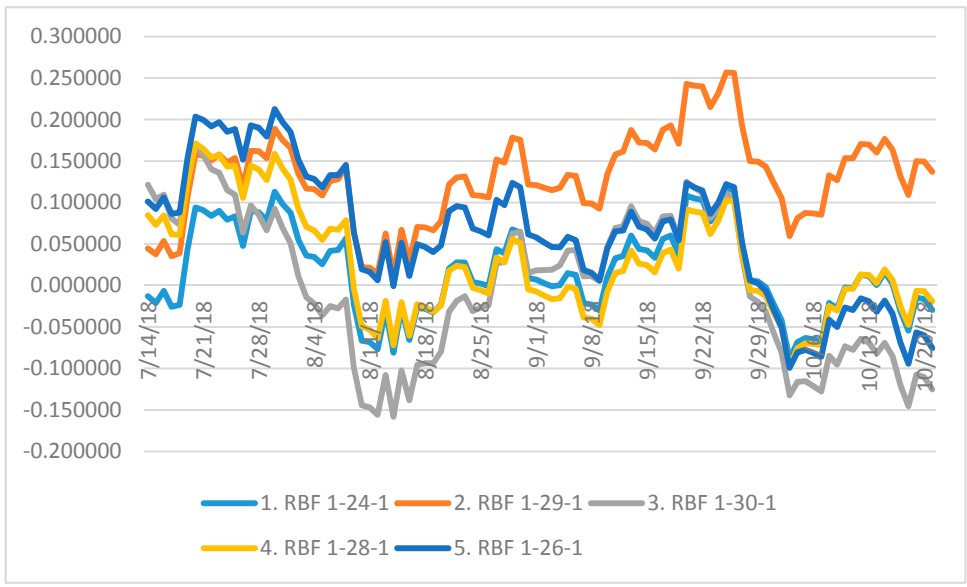

**Figure 6.** Development of equalized time series residuals between 14th July and 21st October 2018. (Source: Own processing.)

It results from the graph that in the monitored period, the sum of the residuals approaching zero appears at all neural networks except for 2. RBF 1-29-1. In this case, all residuals achieve relatively high values. To illustrate the situation, Table 5 shows the sum of the residuals of all equalized time series.

**Table 5.** Sum of residuals of individual equalized time series.

| Statistics | 1. RBF 1-24-1 | 2. RBF 1-29-1 | 3. RBF 1-30-1 | 4. RBF 1-28-1 | 5. RBF 1-26-1 |
|---|---|---|---|---|---|
| Sum of residuals | 10.753689 | 1.369654 | 10.598169 | 4.290912 | 0.676559 |

Source: Own processing.

When leaving aside the residuals fluctuations during the whole monitored period, the sum of the residuals will be zero in the ideal case. The network closest to the ideal case is 5. RBF 1-26-1 network, whose sum of the residuals is nearly 0.676. The highest sum of the residuals is achieved by 1.RBF 1-24-1, and 3. RBF 1-30-1, whose sum of the residuals are above 10 in both cases. The results of the last analysis show that the most successful time series is 5. RBF 1-26-1 network. However, it shall be noted that, considering the 3303 measurements carried out, even the value of 10 appears minimal. We can, therefore, claim that 1. RBF 1-24-1 network remains the most successful neural structure.

*4.2. Neural Structures B*

Based on the procedure mentioned above, 100,000 neural networks were generated, out of which 5 with the best characteristics were retained. Their overview is shown in Table 6.

**Table 6.** Retained neural networks.

| Index | Network | Training Perform. | Testing Perform. | Validation Perform. | Training Error | Testing Error | Validation Error | Training Algorithm | Error Function | Activation of Hidden Layer | Output Activation Function |
|---|---|---|---|---|---|---|---|---|---|---|---|
| 1 | MLP 61-10-1 | 0.998390 | 0.997106 | 0.996723 | 0.000924 | 0.001732 | 0.001931 | BFGS (Quasi-Newton) 419 | Sum.sq. | Tanh | Identity |
| 2 | MLP 61-11-1 | 0.998491 | 0.996916 | 0.996710 | 0.000866 | 0.001831 | 0.001958 | BFGS (Quasi-Newton) 468 | Sum.sq. | Tanh | Tanh |
| 3 | MLP 61-11-1 | 0.998505 | 0.997400 | 0.996715 | 0.000858 | 0.001549 | 0.001944 | BFGS (Quasi-Newton) 475 | Sum.sq. | Tanh | Sinus |
| 4 | MLP 61-11-1 | 0.998490 | 0.997074 | 0.996715 | 0.000866 | 0.001744 | 0.001939 | BFGS (Quasi-Newton) 394 | Sum.sq. | Tanh | Identity |
| 5 | MLP 61-10-1 | 0.998250 | 0.996974 | 0.996990 | 0.001004 | 0.001799 | 0.001801 | BFGS (Quasi-Newton) 398 | Sum.sq. | Tanh | Logistic |

Source: Own processing.

These are only multi-layer perceptron networks. The input layer contains four variables: time, year, month, and day. Time is represented by one neuron in the input layer, year by 10 neurons, day in a week by 7 neurons, and day in a month by 31 neurons in the input layer of the networks. The input layers of the generated and retained neural networks thus consist of 61 neurons. The hidden layers contain 10 or 11 neurons. The output layer logically contains only one neuron and one output variable—Euro exchange rate to Yuan. For all networks, Quasi-Newton training algorithm was applied, either in version 3 or 4. All neural networks used the same function for the activation of the neurons in the hidden layer, namely hyperbolic tangent. For the activation of the output layer, the retained neural networks used the following functions: hyperbolic tangent, logistic function, sinus, and function Identity (for more details, see Table 6).

We are looking for such a network whose performance in all data sets is ideally the same (here, it should be noted that the division of the data into the data sets was random). The error shall be as small as possible.

The performance of the individual data sets is given in the form of the correlation coefficient. The individual data sets values by concrete neural networks are shown in Table 7.

**Table 7.** Correlation coefficients of individual data sets.

| Network | Euro to Chinese Yuan (Training) | Euro to Chinese Yuan (Testing) | Euro to Chinese Yuan (Validation) |
|---|---|---|---|
| 1. MLP 61-10-1 | 0.998390 | 0.997106 | 0.996723 |
| 2. MLP 61-11-1 | 0.998491 | 0.996916 | 0.996710 |
| 3. MLP 61-11-1 | 0.998505 | 0.997400 | 0.996715 |
| 4. MLP 61-11-1 | 0.998490 | 0.997074 | 0.996715 |
| 5. MLP 61-10-1 | 0.998250 | 0.996974 | 0.996990 |

Source: Own processing.

It results from the table that the performance of all retained neural structures is approximately the same. The slight differences do not affect the performance of the individual networks. The correlation coefficient of all training data sets is above 0.998. The correlation coefficient of the testing data sets is nearly 0.997 or above this value. The correlation coefficient of all neural networks validation data sets is above 0.996. At the same time, it has to be noted that the error in all data sets is in the interval of more than 0.0008 to less than 0.002. The error differences of equalized time series in the individual data sets are almost negligible.

To choose the most suitable neural structure, further analysis of the results obtained must be carried out. Table 8 shows the basic statistical characteristics of the individual data sets for all neural structures.

In the case of retained neural structures, the differences of equalized time series are minimal both in terms of absolute values and residuals. Not even on the basis of the data obtained, it is possible to identify unambiguously which of the retained neural structures shows the best results. All the neural structures appear to be applicable in practice.

**Table 8.** Statistics of individual data sets by retained neural structures.

| Statistics | 1. MLP 61-10-1 | 2. MLP 61-11-1 | 3. MLP 61-11-1 | 4. MLP 61-11-1 | 5. MLP 61-10-1 |
|---|---|---|---|---|---|
| Minimal prediction (training) | 6.59204 | 6.60787 | 6.61005 | 6.59059 | 6.65057 |
| Maximal prediction (training) | 10.25774 | 10.20064 | 10.25455 | 10.28336 | 10.22089 |
| Minimal prediction (testing) | 6.62588 | 6.63547 | 6.63945 | 6.64194 | 6.64934 |
| Maximal prediction (testing) | 10.32071 | 10.19929 | 10.23666 | 10.24191 | 10.21010 |
| Minimal prediction (validation) | 6.57746 | 6.68662 | 6.67522 | 6.62117 | 6.66080 |
| Maximal prediction (validation) | 10.23630 | 10.19925 | 10.23854 | 10.28137 | 10.20854 |
| Minimal residuals (training) | −0.16067 | −0.18343 | −0.15628 | −0.16430 | −0.19135 |
| Maximal residuals (training) | 0.39922 | 0.49966 | 0.49448 | 0.45441 | 0.46364 |
| Minimal residuals (testing) | −0.25721 | −0.22189 | −0.18788 | −0.23063 | −0.18800 |
| Maximal residua (testing) | 0.19605 | 0.22268 | 0.21866 | 0.18687 | 0.29676 |
| Minimal residuals (validation) | −0.22231 | −0.16419 | −0.16207 | −0.20030 | −0.17103 |
| Maximal residuals (validation) | 0.55087 | 0.53214 | 0.56542 | 0.53715 | 0.48879 |
| Minimal standard residuals (training) | −5.28571 | −6.23421 | −5.33626 | −5.58204 | −6.03872 |
| Maximal standard residuals (training) | 13.13338 | 16.98175 | 16.88430 | 15.43838 | 14.63152 |
| Minimal standard residuals (testing) | −6.18008 | −5.18477 | −4.77384 | −5.52315 | −4.43254 |
| Maximal standard residuals (testing) | 4.71053 | 5.20318 | 5.55595 | 4.47499 | 6.99679 |
| Minimal standard residuals (validation) | −5.05836 | −3.71056 | −3.67572 | −4.54863 | −4.03019 |
| Maximal standard residuals (validation) | 12.53453 | 12.02629 | 12.82361 | 12.19844 | 11.51780 |

Source: Own processing.

Figure 7 shows a line graph presenting the actual development of the Euro to Yuan exchange rate and also the development of predictions by means of the individual generated and retained networks (or equalized time series). The blue curve follows the actual development of the exchange rate, while the other curves always follow one of the retained neural networks.

It is evident from the graph that all the neural networks predict the development of the Euro to Yuan exchange rate in the individual intervals almost identically. Also, the course of the equalized time series is very similar to the actual course of the Euro to Chinese Yuan exchange rate. They follow the gradient of the curve representing the development of the Euro to Yuan exchange rate, but at the same time, they also precisely describe the extremes of the curve.

Given that the graph in Figure 7 includes 3303 data on Euro exchange rate to Yuan, it may appear unclear. Therefore, it would be good to demonstrate the situation on the selected data interval. The graph in Figure 8 shows a comparison of the actual development of the Euro to Yuan exchange rate in the last 100 days of the monitored period, i.e., from 14th July to 21st October 2018.

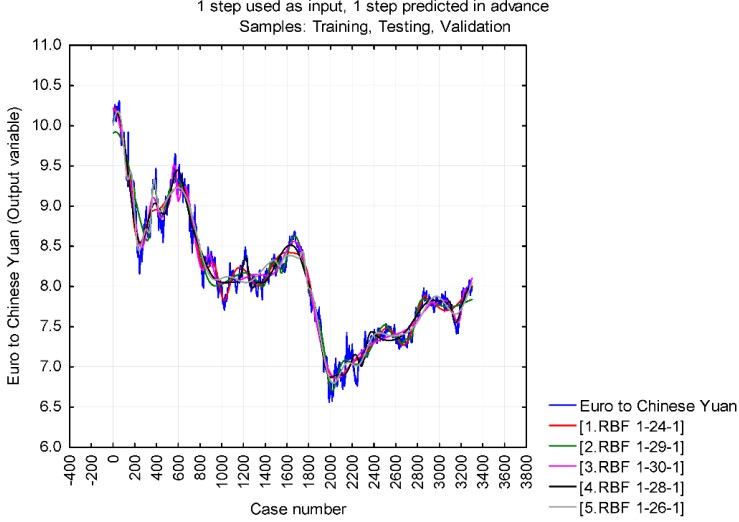

**Figure 7.** Line graph—development of Euro exchange rate to Yuan predicted by neural networks in comparison with the actual exchange rate in the monitored period. (Source: Own processing.)

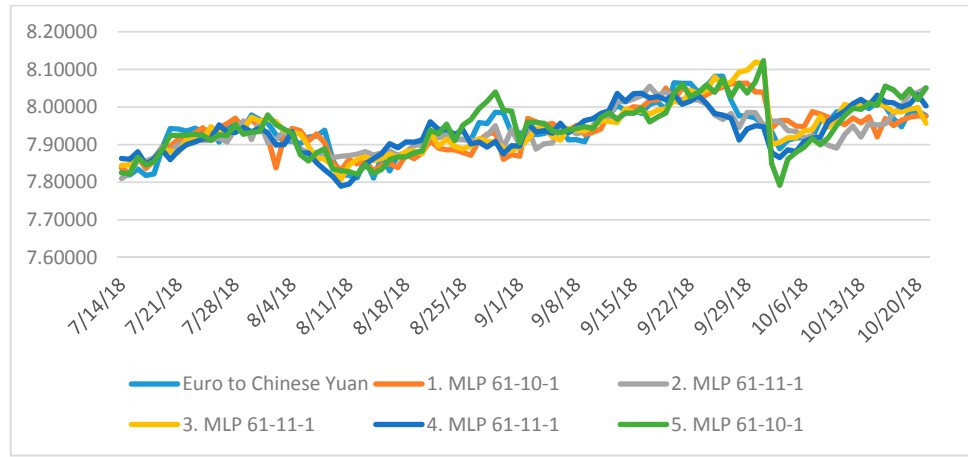

**Figure 8.** Line graph—development of Euro to Yuan predicted by neural networks compared to actual exchange rate between 14th July and 21st October 2018. (Source: Own processing.)

The graph shows that all neural networks can imitate the exchange rate of Euro to Yuan quite well. The deviation in the individual interval sections is 0.05 Yuan at most. Bigger differences were detected only on 25th September 2018, 1st October 2018, and 2nd October 2018; however, even on these days, the differences are not bigger than 0.1 Yuan. Based on the graph comparison, it can be stated that all the retained neural networks are applicable for the prediction. Therefore, examining residuals appears to be interesting. The development of the residuals between 14th July and 21st October 2018 is seen in Figure 9.

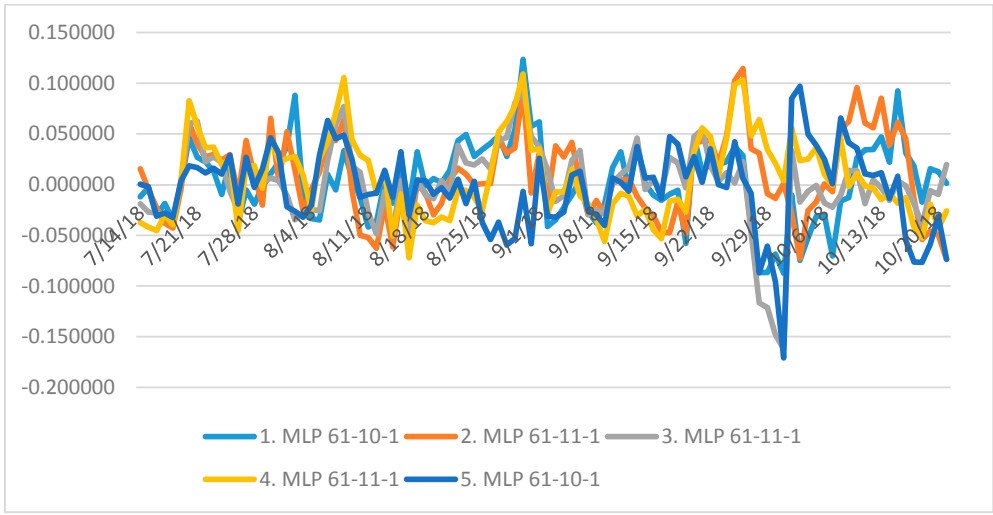

**Figure 9.** Development of residuals of equalized time series between 14th July and 21st October 2018. (Source: Own processing.)

It results from the graph that the sum of the residuals approaching zero can be seen with all neural networks in the monitored period. To illustrate the situation better, Table 9 shows the sum of the residuals of all equalized time series for the entire period.

**Table 9.** Sum of residuals of individual equalized time series.

| Statistics | 1. MLP 61-10-1 | 2. MLP 61-11-1 | 3. MLP 61-11-1 | 4. MLP 61-11-1 | 5. MLP 61-10-1 |
|---|---|---|---|---|---|
| Sum of residuals | −0.677755 | 0.636444 | −0.584423 | 0.844007 | 0.000112 |

Source: Own processing.

Leaving aside the individual fluctuations of the residuals during the entire monitored period, the sum of the residuals is zero in the ideal case. Closer to zero is the sum of the residuals of the 5th neural network 5. MLP 61-10-1, which is 0.0001. On the contrary, the highest value of the sum is achieved by 4. MLP 61-11-1 network (0.844). The differences are minimal. It is therefore possible to confirm that all the retained neural networks are capable of equalizing the time series of Euro to Chinese Yuan exchange rate reliably and can be used for a machine prediction of these two currencies' exchange rate.

*4.3. Comparising Results of A and B*

All the generated and retained artificial neural networks were able to equalize the examined time series—Euro to Yuan exchange rate. Even the comparison of the correlation coefficients (see Tables 3 and 7) clearly shows a higher performance of the B alternative, that is, the MLP neural networks (when including additional categorical variables). This is also reflected in assessing the basic predictions statistics of the equalized time series (see Tables 4 and 8). The retained MLP networks, or their equalized time series, show smaller mutual differences in training, testing, and validation data sets than in the case of the RBF networks (without additional variable). This can be seen in Figures 4–10. At first sight, it is clear that only the MLP networks of the B alternative are capable of capturing the actual time series course (for more details, see Figure 10).

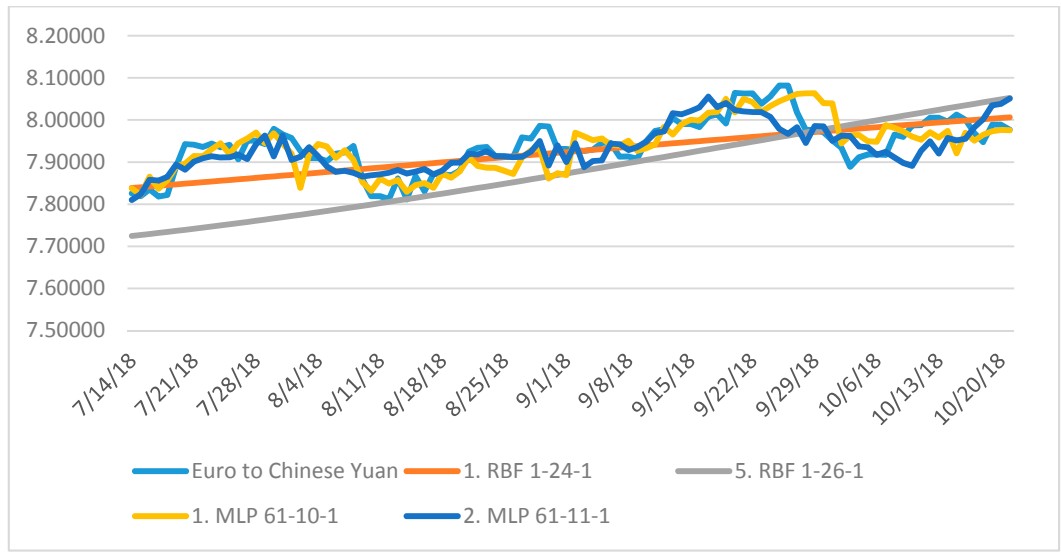

**Figure 10.** Line graph—comparison of retained neural networks in equalizing time series of development of Euro to Yuan exchange rate between 14th July and 21st October 2018. (Source: Own processing.)

The seasonality of both currencies' mutual exchange rate development was observed in three aspects. The information whether the seasonality in time series, or equalized time series, can be identified is given by the sum of absolute residuals. First, we focused on the differences between the individual months (for more details, see Table 10).

The table shows the differences between the individual months. Specifically, e.g., June showed higher absolute residual values (volatility). Conversely, the lowest absolute residual values were recorded in November.

**Table 10.** Equalized time series residuals—months in a year.

| Identification | 1. MLP 61-10-1 | 2. MLP 61-11-1 | 3. MLP 61-11-1 | 4. MLP 61-11-1 | 5. MLP 61-10-1 |
|---|---|---|---|---|---|
| January | 11.049 | 10.823 | 10.760 | 9.576 | 10.583 |
| February | 10.208 | 10.706 | 10.649 | 10.921 | 10.558 |
| March | 9.355 | 8.753 | 9.338 | 10.224 | 10.468 |
| April | 9.499 | 9.986 | 9.890 | 8.145 | 10.492 |
| May | 11.698 | 11.565 | 10.245 | 9.952 | 10.897 |
| June | 11.160 | 10.691 | 9.916 | 10.696 | 10.429 |
| July | 10.346 | 9.732 | 9.475 | 9.349 | 9.144 |
| August | 10.046 | 9.895 | 9.921 | 10.141 | 10.288 |
| September | 9.110 | 8.090 | 8.765 | 10.172 | 8.540 |
| October | 10.895 | 9.889 | 9.385 | 11.000 | 11.284 |
| November | 9.608 | 9.249 | 8.992 | 9.866 | 9.779 |
| December | 8.495 | 9.717 | 9.073 | 9.373 | 10.747 |
| Minimum | 8.495 | 8.090 | 8.765 | 8.145 | 8.540 |
| Maximum | 11.698 | 11.565 | 10.760 | 11.000 | 11.284 |

Source: Own processing.

Volatility was also monitored at the level of one month, in particular by the individual days of a month (see Table 11).

**Table 11.** Equalized time series absolute residuals—days of a month.

| Identification | 1. MLP 61-10-1 | 2. MLP 61-11-1 | 3. MLP 61-11-1 | 4. MLP 61-11-1 | 5. MLP 61-10-1 |
|---|---|---|---|---|---|
| 1 | 5.927 | 5.566 | 4.828 | 5.766 | 5.784 |
| 2 | 5.199 | 5.490 | 5.185 | 5.099 | 5.149 |
| 3 | 4.806 | 4.281 | 4.438 | 4.469 | 4.152 |
| 4 | 3.970 | 3.610 | 4.044 | 3.686 | 3.654 |
| 5 | 3.746 | 3.189 | 3.515 | 4.268 | 3.457 |
| 6 | 3.402 | 3.321 | 3.412 | 3.828 | 3.342 |
| 7 | 3.606 | 3.758 | 3.603 | 4.113 | 3.950 |
| 8 | 3.611 | 3.770 | 3.596 | 3.731 | 3.511 |
| 9 | 3.740 | 4.015 | 3.782 | 3.447 | 4.004 |
| 10 | 4.066 | 3.742 | 4.040 | 3.495 | 4.225 |
| 11 | 4.102 | 3.579 | 4.245 | 3.702 | 4.096 |
| 12 | 3.638 | 3.511 | 3.583 | 4.091 | 3.923 |
| 13 | 3.890 | 3.806 | 3.702 | 3.743 | 4.264 |
| 14 | 4.071 | 4.305 | 3.853 | 4.373 | 4.403 |
| 15 | 4.327 | 4.281 | 3.050 | 4.186 | 4.111 |
| 16 | 3.915 | 3.924 | 3.119 | 3.792 | 3.957 |
| 17 | 3.749 | 3.760 | 3.268 | 3.489 | 3.924 |
| 18 | 3.812 | 3.883 | 3.710 | 3.841 | 3.747 |
| 19 | 3.925 | 3.974 | 3.923 | 3.840 | 3.937 |
| 20 | 4.502 | 3.790 | 4.516 | 4.120 | 4.211 |
| 21 | 4.178 | 3.781 | 4.432 | 4.134 | 4.089 |
| 22 | 3.519 | 3.460 | 3.663 | 3.393 | 3.720 |
| 23 | 3.127 | 3.339 | 3.330 | 3.400 | 3.390 |
| 24 | 3.246 | 3.500 | 3.323 | 3.213 | 3.525 |
| 25 | 3.142 | 3.765 | 3.603 | 3.611 | 3.665 |
| 26 | 3.228 | 3.366 | 3.337 | 3.570 | 3.673 |
| 27 | 3.365 | 3.252 | 3.376 | 3.106 | 3.817 |
| 28 | 3.796 | 3.823 | 3.667 | 3.685 | 4.272 |
| 29 | 3.855 | 3.893 | 3.586 | 3.253 | 4.068 |
| 30 | 4.729 | 4.690 | 4.167 | 4.148 | 4.377 |
| 31 | 3.278 | 2.671 | 2.514 | 2.821 | 2.814 |
| Minimum | 3.127 | 2.671 | 2.514 | 2.821 | 2.814 |
| Maximum | 5.927 | 5.566 | 5.185 | 5.766 | 5.784 |

Source: Own processing.

Table 11 shows that even a day in a month can play a role. Specifically, the first few days of the month shows higher volatility. Conversely, the lowest volatility was identified in the last few days

of the month. However, this value can be influenced by the fact that not all months have 31 days. Therefore, the sum of the absolute residuals on the 29th, 30th, and 31st day is lower than that in the previous days. However, the volatility is higher during the first few days of the month.

Finally, possible differences between the individual days of a week were studied (see Table 12).

**Table 12.** Equalized time series absolute residuals—days of a week.

| Identification | 1. MLP 61-10-1 | 2. MLP 61-11-1 | 3. MLP 61-11-1 | 4. MLP 61-11-1 | 5. MLP 61-10-1 |
|---|---|---|---|---|---|
| Monday | 15.837 | 15.551 | 16.293 | 16.503 | 16.774 |
| Tuesday | 16.647 | 17.015 | 16.715 | 17.139 | 17.460 |
| Wednesday | 18.542 | 17.779 | 16.550 | 17.462 | 18.049 |
| Thursday | 17.204 | 17.463 | 16.446 | 17.196 | 18.050 |
| Friday | 18.093 | 17.418 | 16.697 | 17.139 | 17.271 |
| Saturday | 17.244 | 16.603 | 16.491 | 17.022 | 17.487 |
| Sunday | 17.903 | 17.266 | 17.218 | 16.953 | 18.119 |
| Minimum | 15.837 | 15.551 | 16.293 | 16.503 | 16.774 |
| Maximum | 18.542 | 17.779 | 17.218 | 17.462 | 18.119 |

Source: Own processing.

The table shows lower volatility only on Monday. The volatility in the other days is approximately the same.

To sum up, seasonality is visible in terms of a calendar month, a day of a month, and a day of a week (albeit only slightly in some cases). Thanks to the selected method of equalizing the time series (or adding another variable) and prediction of the future time series development, more accurate results were obtained, which can be better used in practice.

Also, thanks to additional categorical variables, the researchers succeeded in identifying the seasonal effect on the exchange rate, and the result is definitely more accurate than in the case of predicting the future development based on a single continuous variable—time.

However, if we include seasonal variables, we get a better result.

It should be noted that neural networks alone do not address seasonality. The difference between the two calculation options is obvious. Previous publications did not address the inclusion of seasonality in models; this is the added value of this article.

## 5. Discussion and Conclusions

The objective of the contribution was to propose a methodology for considering seasonal fluctuations when equalizing time series by means of artificial neural networks on the example of Euro and Chinese Yuan.

Generally, each prediction was given by a certain degree of probability with which it gets fulfilled. Predicting the future development of any variable means to estimate its future development based on the data from the previous periods. Although we are able to include most of the factors affecting the target variable in the model, there is always a certain simplification of reality and we thus work with a certain degree of probability that the predicted scenario will become true.

In the contribution, the authors focused on comparing the application of the same tool at a different initial assignment. Despite it had seemed before the experiment that there was no reason for including the categorical variable in order to capture the seasonal fluctuations of the two currencies exchange rates, the opposite proved to be true. The additional variables in the form of year, month, day in a month, and day in a week in which the value had been measured resulted in higher accuracy and order of the time series.

The development of the two currencies exchange rate can be predicted based on the statistical methods, causal methods, and intuitive methods. In this contribution, the authors focused on comparing the statistical methods. However, these only provided a possible framework of the monitored variable development. What is important is to work subsequently with the information on

the possible future development of the economic, political, or legal environment. If it is possible to predict their development, it can subsequently be incorporated in the monitored variable. Here, the personality of the evaluators starts to play a role—economists, who, based on their knowledge and experience correct the price set by statistical methods and specified based on the causal links. However, in this case, it appears to be possible to test the prediction only by means of the B alternative, which gives a relatively higher degree of accuracy. Interesting results were reported by the MLP networks.

The objective of the contribution was achieved.

An interesting fact is that in the case of the A alternative, the most successful networks appeared to be the radial basis neural networks. On the contrary, in the case of the B alternative, the most successful ones were the multi-layer perceptron neural networks. It would undoubtedly be an interesting experiment if only one type of neural structures was generated for the given situation, different from the results already obtained (that is, MLP networks for the A alternative, and RBF for the B alternative). The numbers in this article explain the trend. The variable was explained, but also predicted.

**Author Contributions:** Conceptualization, M.V. and J.H.; methodology, M.V.; software, M.V. and J.H.; validation, M.V., J.H., and P.Š.; formal analysis, P.Š.; resources, J.H. and P.Š.; writing—original draft preparation, M.V.; writing—review and editing, J.H.; supervision, M.V.

**Funding:** This research received no external funding.

**Conflicts of Interest:** The authors declare no conflict of interest.

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
