# Peer review of "Equalizing Seasonal Time Series Using Artificial Neural Networks in Predicting the Euro–Yuan Exchange Rate"

_jrfm, doi:10.3390/jrfm12020076_

Round 1
Reviewer 1 Report
This work presents an analysis of seasonal fluctuations to predict the evolution of Euro and Chinese exchange rate. This question is very important to obtain a model that allows to reduce the error predictions. Here are some questions and observations, maybe to improve the work.
1) It should be necessary a higher description of the neural network model. Perhaps, with some figure it's enough for better understandig by the reader. Additionally, a basic references about the mlp neural network should be consider.
2) Has been considered others neural networks like recurrent neural networks or evolino neural networks?
3) How does the different volatility period affect to the seasonal fluctuations time series in the model?
Author Response
Dear Reviewer,
thank you for your comments. You will find everything you need in the attachment.
With kind regards
authors

Reviewer 2 Report
This is an interesting study of the Euro/ Yuan exchange rate using neural networks to forecast this exchange rate. The study finds that including categorical variables for the seasonal effects improves the predictive performance. I have the following comments:
- The introduction could be more specific in what the contribution of this study is to the literature, in terms of data and technique used.
- The literature review could add more studies specifically assessing the exchange rate using neural networks, such as Nag and Mitra (2002).
- At the end of the introduction, there should be a short piece outlining the rest of the paper.
- In the methodology section, a brief explanation of how the neural network used here work could be added.
- It would be useful if an explanation of why exchange rates would be more predictable when accounting for seasonal effects was added, do any other studies find that the exchange rate changes with the seasons?
- The study refers to predicting the exchange rate using neural networks, do they mean explain the exchange rate rather than predict? If it is predict, are these out-of-sample? Also I think more explanation of the type of predictions would need to be added, for instance can they be measured using prediction errors?
Minor points
- The paper could do with a final proof reading.
- In case 1 the independent variable is time, is this a time trend?
- More explanation of the seasonal dummy variables needs to be added.
References
Nag, A. and A. Mitra, (2002). Forecasting daily foreign exchange rates using genetically optimised neural networks, Journal of Forecasting, 21, 501-511.
Author Response

(The authors gave the same response as above.)

Round 2
Reviewer 2 Report
The revised version of this paper addresses the earlier comments. A final couple of points:
- The first sentence of the abstract would be better with: ' The exchange rate is one of the most...'.
- The first sentence in the introduction it may be clearer to say if the authors like: 'In this article, the authors focus on comparing the application of neural networks to the exchange rate using different modelling approaches.
Author Response
Dear Reviewer,
All the required changes have been incorporated.
Thank you for cooperation!